# GRPO IS SECRETLY A PROCESS REWARD MODEL

## ABSTRACT

We prove theoretically that the GRPO RL algorithm induces a non-trivial process reward model (PRM), under certain assumptions regarding within-group overlap of token sequences across completions. We then show empirically that these assumptions are met under real-world conditions: GRPO does in fact induce a non-trivial PRM. Leveraging the framework of GRPO-as-a-PRM, we identify a flaw in the GRPO objective: non-uniformly distributed process steps hinder both exploration and exploitation (under different conditions). We propose a simple modification to the algorithm to mitigate this defect ($\lambda$-GRPO), and show that LLMs trained with $\lambda$-GRPO achieve higher validation accuracy and performance on downstream reasoning tasks—and reach peak performance more rapidly—than LLMs trained with standard GRPO. Our results call into question the advantage of costly, explicitly-defined PRMs for GRPO: we show that it is possible to instead leverage the hidden, built-in PRM structure within the vanilla GRPO algorithm to boost model performance with a negligible impact on training time and cost.

## 1 INTRODUCTION

Process reward models (PRMs)—models that assign reward to intermediate steps (see Section 2.2)—allow for finer-grained reward assignment than outcome-level signals, thereby yielding improved multi-step reasoning performance (Lightman et al., 2024). PRMs are therefore particularly applicable to RL training for step-by-step processes such as mathematical reasoning.

However, training neural PRMs requires costly, step-level human annotation (Zhang et al., 2025), and such models are particularly susceptible to reward hacking (Cui et al., 2025). These shortcomings have resulted in the limited adoption of learned PRMs for RL training Setlur et al. (2025), leading to the development of Monte-Carlo-based and other heuristic, non-neural PRMs (e.g. Wang et al., 2024; Kazemnejad et al., 2025; Hou et al., 2025).

PRMs are typically employed with RL algorithms such as Proximal Policy Optimization (PPO; Schulman et al., 2017) that employ a critic model and/or generalized advantage estimation (GAE). Although Group Relative Policy Optimization (GRPO; Shao et al., 2024) greatly simplifies and reduces the memory consumption of RL training, leading to its adoption for a wide range of applications—e.g. tool use (Qian et al., 2025; Sullivan et al., 2025), RLHF (Yang et al., 2025b), and, in particular, mathematical reasoning (Shao et al., 2024; DeepSeek-AI, 2025)—it does so by eliminating the critic model and GAE of PPO (see Section 2.1). GRPO has therefore not been widely used with PRMs: to the best of our knowledge, Shao et al. (2024), Yang et al. (2025a), and Feng et al. (2025) are the only instances in which GRPO is employed with step-level rewards—and these approaches necessitate the modification of the algorithm to accommodate finer-grained reward signals.

In this paper, we show that—under certain mild assumptions—GRPO induces a Monte-Carlo-based PRM (Section 3). Specifically, we prove theoretically that GRPO assigns rewards (and advantages) derived from outcome-level rewards and Monte-Carlo-sampled completions to sub-trajectories, whenever subsets of trajectories within each group share identical prefixes. We then show empirically that this identical-prefix condition is almost always met under real-world conditions, yielding rich step-level process reward structures. These two findings definitively demonstrate that the GRPO objective covertly assigns and optimizes for complex, structured step-level rewards and advantages.

An investigation of the properties of GRPO's hidden PRM reveals a defect in the objective function that hinders both exploration and exploitation (under different conditions) during RL training (Section 4): namely, a vulnerability to non-uniformly-distributed process steps within a group. To

mitigate this shortcoming, we propose including a PRM-aware normalization factor into the GRPO loss function ($\lambda$-GRPO).

We show that $\lambda$-GRPO results in higher validation accuracy and a $\sim$2x training speedup over standard GRPO (Section 5). On downstream reasoning benchmarks, $\lambda$-GRPO consistently improves over standard GRPO, demonstrating the superiority of our method.

Our findings call into question the utility of dedicated PRMs for GRPO, and suggest that future work may benefit instead from exploiting the step-level reward signal already available to the GRPO algorithm.

## 2 BACKGROUND

### 2.1 GRPO

GRPO is a variant of PPO that discards the critic model and GAE of the latter. Instead, for each query $q$ in the training set, GRPO nondeterministically samples a *group* $\mathbb{G}$ of $k$ trajectories (completions to $q$) and computes the advantage $a_i$ for the completion $g^{(i)} \in \mathbb{G}$ relative to the mean (outcome) reward of $\mathbb{G}$, as in Equation 1 (where $r_i$ is the reward for $g^{(i)}$).

$$a_i = \frac{r_i - r_{mean}(\mathbb{G})}{r_{std}(\mathbb{G})} \tag{1}$$

In our theoretical analysis in Section 3, we make two key assumptions: first, we assume the use of the DAPO token-level policy gradient objective (Yu et al., 2025), rather than sample-level loss. Although it differs from the original GRPO formulation laid out in Shao et al. (2024), Yu et al. (2025) show that this objective leads to more stable training, and it is the standard GRPO loss function employed in commonly-used RL packages (e.g. the TRL GRPO trainer[1]).

Second, we assume that the number of update iterations per batch ($\mu$) is set to $\mu = 1$. Under this assumption, the ratio $P_{i,t}$ is fixed at $1.0$ (see Equation 2b), allowing us to ignore the clipping factor of the GRPO loss function in our theoretical analysis.

Under these two assumptions, the per-group GRPO loss $L_{\text{GRPO}}(\mathbb{G})$ reduces to that in Equation 2.

$$L_{\text{GRPO}}(\mathbb{G}) = \frac{1}{\sum_{g^{(i)} \in \mathbb{G}} len(g^{(i)})} \sum_{g^{(i)} \in \mathbb{G}} \sum_{t=0}^{len(g^{(i)})-1} (P_{i,t} \cdot a_i) - D_{i,t} \tag{2a}$$

$$P_{i,t} = \frac{\pi_\theta(g_t^{(i)} \mid q, g_{:t}^{(i)})}{\pi_{\theta_{old}}(g_t^{(i)} \mid q, g_{:t}^{(i)})} \tag{2b}$$

$$D_{i,t} = \beta \cdot \left( \frac{\pi_{\theta_{ref}}(g_t^{(i)} \mid q, g_{:t}^{(i)})}{\pi_\theta(g_t^{(i)} \mid q, g_{:t}^{(i)})} - ln\frac{\pi_{\theta_{ref}}(g_t^{(i)} \mid q, g_{:t}^{(i)})}{\pi_\theta(g_t^{(i)} \mid q, g_{:t}^{(i)})} - 1 \right) \tag{2c}$$

### 2.2 PROCESS REWARD MODELS (PRMs)

Given an alphabet $\Sigma$ (i.e. set of tokens), we formally define a PRM as a function $f_\phi \colon \Sigma^* \to (\Sigma^* \times \mathbb{R})^*$ parameterized by $\phi$ that maps a trajectory $g \in \Sigma^*$ to the sequence $f_\phi(g) = ((g_{:i_1}, r_0^{(g)}), (g_{i_1:i_2}, r_1^{(g)}), \ldots, (g_{i_{n-1}:}, r_{n-1}^{(g)}))$ of pairs of *process steps* (sub-trajectories) $g_{i_k:i_{k+1}}$ and step-level rewards $r_k^{(g)}$.

---

[1]https://huggingface.co/docs/trl/main/en/grpo_trainer

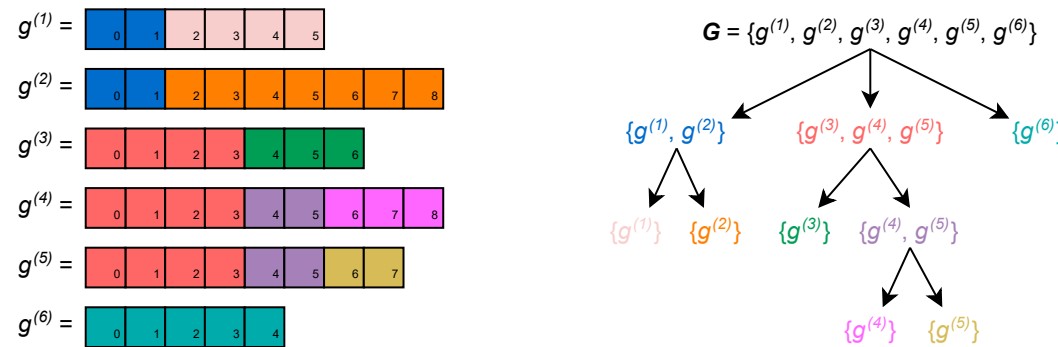

Figure 1: Toy example of a group $\mathbb{G} = \{g^{(1)}, \ldots, g^{(6)}\}$ (left) and its corresponding $\mathcal{B}(\mathbb{G})$ tree (right). Tokens (boxes) are numbered for readability—subscripted numbers within boxes only indicate position. Each process *set* (node in the $\mathcal{B}(\mathbb{G})$ tree) is a set of trajectories that share a common prefix, and corresponds to a process *step* (subtrajectory) spanning those shared tokens: in this figure, colored nodes in $\mathcal{B}(\mathbb{G})$ correspond to those subsequences in $\mathbb{G}$ that span tokens/boxes of the same color[2]. GRPO implicitly assigns a step-level reward and advantage to the tokens of each process step, which are computed as functions of the mean outcome-level reward of each trajectory in the corresponding process set.

While PRMs are typically contrasted with *outcome* reward models (ORMs)—which assign a single reward to the entire trajectory—under the above definition, an ORM $f'_{\phi'}$ is simply a *trivial* PRM: i.e. $f'_{\phi'}(g) = ((g, r_0^{(g)}))$.

Both the division of the trajectory $g$ into steps and the assignment of rewards to those steps are dependent upon the PRM in question. When trajectories are clearly delineated into individual steps—e.g. via ReACT-style prompting (Yao et al., 2023) or instructing the model to divide its reasoning into demarcated steps—the PRM can simply be directed to assign a reward to each pre-defined step (e.g. Li & Li, 2025). In other cases, trajectories are heuristically split into steps—for example, at high-entropy tokens (e.g. Hou et al., 2025).

Although the assignment of step-level reward can be performed by a model with learned parameters $\phi$ (e.g. Uesato et al., 2022), Kazemnejad et al. (2025) and Hou et al. (2025) combine Monte Carlo estimation with outcome-level rewards to yield heuristic PRMs that do not require the labor-intensive annotation of—and are less susceptible to reward-hacking than—their learned counterparts. In cases such as these in which the PRM $f_\phi$ is not learned, we simply consider $\phi$ to be fixed/trivial.

## 3 GRPO'S HIDDEN PRM

In Section 3.1, we prove that GRPO theoretically induces a PRM (given the assumptions of Section 2.1) as defined in Section 2.2. However, this PRM is only non-trivial—i.e. not equivalent to an ORM—if subsets of trajectories within each group share identical initial sub-trajectories.

In Section 3.2, we empirically demonstrate that such rich, overlapping prefix structures arise very frequently under real-world conditions: this shows that GRPO is "secretly" a non-trivial PRM.

### 3.1 THEORETICAL ANALYSIS

Let $\mathcal{B}(\mathbb{G}) = \{\lambda \subseteq \mathbb{G} \mid \exists n \geq 0 \forall g^{(i)}, g^{(k)} \in \lambda : g_{:n}^{(i)} = g_{:n}^{(k)}\}$ be the set of all *process sets*: sets $\lambda \subseteq \mathbb{G}$ of completions such that all $g^{(i)} \in \lambda$ are identical up to the $n^{th}$ token, for some $n \geq 0$ (see Figure 1). Note that there is a natural tree structure on $\mathcal{B}(\mathbb{G})$, which is induced by the $\supseteq$ relation.

---

[2]Same-colored boxes in $\mathbb{G}$ indicate identical *sequences* of tokens across trajectories only: for example, $g_{:4}^{(3)} = g_{:4}^{(4)}$ and $g_{4:6}^{(4)} = g_{4:6}^{(5)}$, but it is not necessarily the case that e.g. $g_0^{(3)} = g_1^{(3)}$.

Each $\lambda \in \mathcal{B}(\mathbb{G})$ defines a process step within each $g^{(i)} \in \lambda$, spanning the subsequence $g^{(i)}_{s(\lambda):e(\lambda)}$ from the $s(\lambda)^{th}$ to the $e(\lambda)^{th}$ tokens of $g^{(i)}$. The endpoint $e(\lambda)$ is defined as the largest $n$ such that $g^{(i)}_{:n} = g^{(k)}_{:n}$ for all $g^{(i)}, g^{(k)} \in \lambda$, and $s(\lambda)$ is defined as the endpoint of the immediate parent $Pa_{\mathcal{B}(\mathbb{G})}(\lambda)$ of $\lambda$ in the tree structure induced on $\mathcal{B}(\mathbb{G})$.

$$e(\lambda) = max\{n \geq 0 \mid \forall g^{(i)}, g^{(k)} \in \lambda : g^{(i)}_{:n} = g^{(k)}_{:n}\}$$

$$s(\lambda) = \begin{cases} 0 & \text{if } \lambda = root(\mathcal{B}(G)) \\ e(Pa_{\mathcal{B}(\mathbb{G})}(\lambda)) & \text{otherwise} \end{cases}$$

For example, $s(\{g^{(4)}, g^{(5)}\}) = e(\{g^{(3)}, g^{(4)}, g^{(5)}\}) = 4$ in Figure 1, and $e(\{g^{(4)}, g^{(5)}\}) = 6$: the process step corresponding to $\{g^{(4)}, g^{(5)}\}$ spans $g^{(4)}_{4:6}$ and $g^{(5)}_{4:6}$.

For each $g^{(i)} \in \mathbb{G}$ and each $0 \leq t < len(g^{(i)})$, let $\lambda^{(i,t)} \in \mathcal{B}(\mathbb{G})$ denote the unique process set such that $g^{(i)} \in \lambda^{(i,t)}$ and $s(\lambda^{(i,t)}) \leq t < e(\lambda^{(i,t)})$. In other words, $\lambda^{(i,t)}$ is the process step to which the token $g^{(i)}_t$ belongs. In Figure 1, $\lambda^{(i,t)}$ corresponds to the set whose color matches that of $g^{(i)}_t$: $\lambda^{(1,0)} = \{g^{(1)}, g^{(2)}\}$, $\lambda^{(1,3)} = \{g^{(1)}\}$, $\lambda^{(5,5)} = \{g^{(4)}, g^{(5)}\}$, etc.

Now, for each process step defined by some $\lambda \in \mathcal{B}(\mathbb{G})$, we define the step-level reward $\hat{R}(\lambda)$ via Monte Carlo estimation (Equation 3): $\hat{R}(\lambda)$ is the mean outcome-level reward of each trajectory in $\lambda$. In other words, $\hat{R}(\lambda)$ is the mean reward of each leaf node dominated by $\lambda$ in the tree structure induced on $\mathcal{B}(\mathbb{G})$—i.e. of each sampled completion to the process step defined by $\lambda$.

$$\hat{R}(\lambda) = \frac{\sum_{g^{(i)} \in \lambda} r_i}{|\lambda|} \tag{3}$$

For each trajectory $g^{(i)} \in \mathbb{G}$ and each $0 \leq t < len(g^{(i)})$ define the reward $R_{i,t}$ for the token $g^{(i)}_t$ as the reward of the process step to which $g^{(i)}_t$ belongs: $R_{i,t} = \hat{R}(\lambda^{(i,t)})$. For example, in Figure 1, the step-level reward for the sub-trajectories $g^{(3)}_{:4}$, $g^{(4)}_{:4}$, $g^{(5)}_{:4}$ is the mean of the outcome-level rewards for $g^{(3)}$, $g^{(4)}$, and $g^{(5)}$: $R_{3,0} = \cdots = R_{5,3} = mean(\{r_3, r_4, r_5\})$.

By the definition given in Section 2.2, $R_{i,t}$ and $\mathcal{B}(\mathbb{G})$ clearly define a PRM: each $g^{(i)} \in \mathbb{G}$ is mapped to the sequence $(g^{(i)}_{:s(\lambda_1)}, R_{i,0}), (g^{(i)}_{s(\lambda_1):s(\lambda_2)}, R_{i,s(\lambda_1)}), \ldots, (g^{(i)}_{s(\lambda_n):}, R_{i,s(\lambda_n)})$, where $(\lambda_0 = \mathbb{G}) \to \cdots \to (\lambda_n = \{g^{(i)}\})$ is the unique path in the tree structure induced on $\mathcal{B}(\mathbb{G})$ from the root $\mathbb{G}$ to the node $\{g^{(i)}\}$.

Now, define the step-level advantage $A_{i,t}$ for the token $g^{(i)}_t$ in an analogous manner to the original GRPO definition in Equation 1—i.e. as the normalized difference between the step-level reward $R_{i,t}$ for $g^{(i)}_t$ and the mean reward of $\mathbb{G}$: $A_{i,t} = (R_{i,t} - r_{mean}(\mathbb{G}))/r_{std}(\mathbb{G})$.

Replacing the term $a_i$ with $A_{i,t}$ in Equation 2a yields a PRM-aware RL objective (Equation 4).

$$L_{\text{PRM}}(\mathbb{G}) = \frac{1}{\sum_{g^{(i)} \in \mathbb{G}} len(g^{(i)})} \sum_{g^{(i)} \in \mathbb{G}} \sum_{t=0}^{len(g^{(i)})-1} (P_{i,t} \cdot A_{i,t}) - D_{i,t} \tag{4}$$

We now show that the standard GRPO objective defined in Equation 2a with outcome-level rewards ($L_{\text{GRPO}}$) is equivalent to the PRM defined in Equations 3-4 ($L_{\text{PRM}}$).

**Theorem 1.** *For any query $q$, policy $\pi_\theta$, and group $\mathbb{G} \sim \pi_\theta(- \mid q)$ with outcome-level rewards $\{r_i\}_{g^{(i)} \in \mathbb{G}}$: $L_{PRM}(\mathbb{G}) = L_{PRM}(\mathbb{G})$.*

*Proof.* Let $\lambda$ be any process set in $\mathcal{B}(\mathbb{G})$, and let $t$ be any integer such that $s(\lambda) \leq t < e(\lambda)$. We first prove that the sum of the PRM loss terms $P_{i,t} \cdot A_{i,t} - D_{i,t}$ for each trajectory $g^{(i)} \in \lambda$ at the token $t$ is equivalent to the sum of the standard GRPO loss terms $P_{i,t} \cdot a_i - D_{i,t}$ (Equation 5).

Recall that for any $g^{(i)}, g^{(k)} \in \lambda$, $g_{:t+1}^{(i)} = g_{:t+1}^{(k)}$ by definition: therefore, $P_{i,t} = P_{k,t}$ and $D_{i,t} = D_{k,t}$ (Equations 2b-2c). Again by definition, $g^{(i)}, g^{(k)} \in \lambda$ implies that $R_{i,t} = R_{k,t} = \hat{R}(\lambda)$ (Equation 3), and so $A_{i,t} = A_{k,t}$. As such, we may define $\hat{P}_t(\lambda) = P_{k,t}$, $\hat{D}_t(\lambda) = D_{k,t}$, and $\hat{A}(\lambda) = A_{k,t}$ in Equation 5, choosing any arbitrary $g^{(k)} \in \lambda$.

$$
\sum_{g^{(i)} \in \lambda} (P_{i,t} \cdot A_{i,t}) - D_{i,t} = |\lambda| \cdot ((\hat{P}_t(\lambda) \cdot \hat{A}(\lambda)) - \hat{D}_t(\lambda))
$$

$$
= |\lambda| \cdot \left( \left( \hat{P}_t(\lambda) \cdot \frac{\hat{R}(\lambda) - r_{mean}(\mathbb{G})}{r_{std}(\mathbb{G})} \right) - \hat{D}_t(\lambda) \right) = |\lambda| \cdot \left( \left( \hat{P}_t(\lambda) \cdot \frac{\sum_{g^{(i)} \in \lambda} \frac{r_i}{|\lambda|} - r_{mean}(\mathbb{G})}{r_{std}(\mathbb{G})} \right) - \hat{D}_t(\lambda) \right)
$$

$$
= \left( \hat{P}_t(\lambda) \frac{|\lambda|(\sum_{g^{(i)} \in \lambda} \frac{r_i}{|\lambda|} - r_{mean}(\mathbb{G}))}{r_{std}(\mathbb{G})} \right) - |\lambda| \hat{D}_t(\lambda) = \left( \hat{P}_t(\lambda) \frac{\sum_{g^{(i)} \in \lambda} r_i - \sum_{g^{(i)} \in \lambda} r_{mean}(\mathbb{G})}{r_{std}(\mathbb{G})} \right) - |\lambda| \hat{D}_t(\lambda)
$$

$$
= \left( \sum_{g^{(i)} \in \lambda} P_{i,t} \frac{r_i - r_{mean}(\mathbb{G})}{r_{std}(\mathbb{G})} \right) - \sum_{g^{(i)} \in \lambda} D_{i,t} = \sum_{g^{(i)} \in \lambda} (P_{i,t} \cdot a_i) - D_{i,t}
$$

$$(5)$$

Now, letting $t_{max} = max_{g^{(i)} \in \mathbb{G}} \, len(g^{(i)})$, for each $0 \leq t < t_{max}$ we can define a partition $\mathbb{X}_t \subseteq \mathcal{B}(\mathbb{G})$ of $\{g^{(i)} \in \mathbb{G} \mid len(g^{(i)}) \leq t\}$ such that $\mathbb{X}_t = \{\lambda \in \mathcal{B}(\mathbb{G}) \mid s(\lambda) \leq t < e(\lambda)\}$ is the set of all process sets corresponding to a token span containing the index $t$. The GRPO loss term $L_{\text{GRPO}}(\mathbb{G})$ (Equation 2a) can be equivalently expressed as in Equation 6 (and analogously for $L_{\text{PRM}}(\mathbb{G})$ of Equation 4).

$$
L_{\text{GRPO}}(\mathbb{G}) = \frac{1}{\sum_{g^{(i)} \in \mathbb{G}} len(g^{(i)})} \cdot \sum_{t=0}^{t_{max}-1} \sum_{\lambda \in \mathbb{X}_t} \sum_{g_i \in \lambda} (P_{i,t} \cdot a_i) - D_{i,t} \qquad (6)
$$

We then have the following equalities by Equations 5 and 6:

$$
L_{\text{GRPO}}(\mathbb{G}) = \frac{1}{\sum_{g^{(i)} \in \mathbb{G}} len(g^{(i)})} \cdot \sum_{g^{(i)} \in \mathbb{G}} \sum_{t=0}^{len(g^{(i)})-1} (P_{i,t} \cdot a_i) - D_{i,t} =
$$

$$
\frac{1}{\sum_{g^{(i)} \in \mathbb{G}} len(g^{(i)})} \cdot \sum_{t=0}^{t_{max}-1} \sum_{\lambda \in \mathbb{X}_t} \sum_{g^{(i)} \in \lambda} (P_{i,t} \cdot a_i) - D_{i,t} = \frac{1}{\sum_{g^{(i)} \in \mathbb{G}} len(g^{(i)})} \cdot \sum_{t=0}^{t_{max}-1} \sum_{\lambda \in \mathbb{X}_t} \sum_{g^{(i)} \in \lambda} (P_{i,t} \cdot A_{i,t}) - D_{i,t}
$$

$$
= \frac{1}{\sum_{g^{(i)} \in \mathbb{G}} len(g^{(i)})} \cdot \sum_{g^{(i)} \in \mathbb{G}} \sum_{t=0}^{len(g^{(i)})-1} (P_{i,t} \cdot A_{i,t}) - D_{i,t} = L_{\text{PRM}}(\mathbb{G})
$$

$\square$

In other words, the standard GRPO objective defined in Equation 2a automatically induces the PRM and PRM-aware objective defined in Equations 3-4.

## 3.2 EMPIRICAL ANALYSIS

The theoretical analysis in Section 3.1 shows that the GRPO objective induces a PRM: it remains to be shown, however, that this induced PRM is *non-trivial*. We refer to the set $\mathcal{B}(\mathbb{G})$ of process sets as *trivial* if it contains only singleton sets[3]—i.e. $\mathcal{B}(\mathbb{G}) = \{\mathbb{G}\} \cup \{\{g^{(i)}\} \mid g^{(i)} \in \mathbb{G}\}$.

---

[3]And, by definition, $\mathbb{G}$ itself, with $s(\mathbb{G}) = e(\mathbb{G}) = 0$.

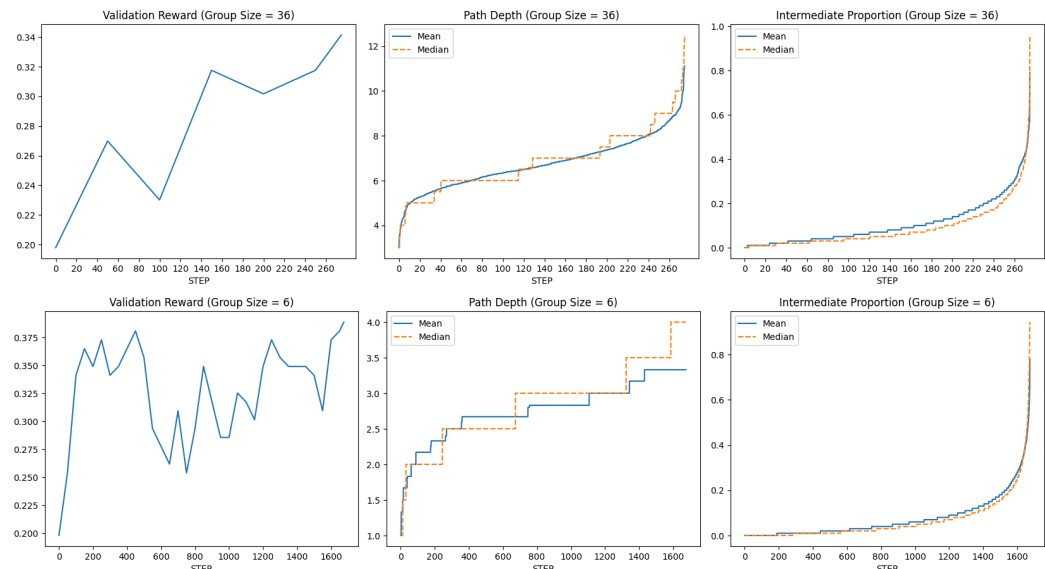

Figure 2: Validation reward (exact-match accuracy; left), $\mathcal{B}(\mathbb{G})$ root-to-terminal path depth (center), and proportions of trajectories spanned by intermediate (non-terminal) process steps (right) for GRPO runs with group sizes of 36 (top) and 6 (bottom).

If $\mathcal{B}(\mathbb{G})$ is trivial, then there are no meaningful process steps within each trajectory, and only the outcome reward has an effect on learning. On the other hand, if $\mathcal{B}(\mathbb{G})$ is *not* trivial, then Theorem 1 entails that the standard GRPO objective of Equation 2a induces a non-trivial PRM.

To analyze the complexity of real-world, GRPO-derived step-level rewards, we empirically evaluated $\mathcal{B}(\mathbb{G})$ structures generated during standard GRPO training: for each group $\mathbb{G}$, we computed its $\mathcal{B}(\mathbb{G})$ tree, and measured the number of intermediate nodes between the root $\mathbb{G}$ and each terminal node $\{g^{(i)}\}$ (*path depth*), as a proxy for the complexity of the $\mathcal{B}(\mathbb{G})$ structure.

In addition, for each completion $g^{(i)} \in \mathbb{G}$, we counted the number of tokens $n_{term}^{(i)} = e(\{g^{(i)}\}) - s(\{g^{(i)}\})$ contained in the process step corresponding to the terminal node $\{g^{(i)}\}$—i.e. the number of tokens unique to $g^{(i)}$—and calculated the *intermediate proportion* $p_i$ of $g^{(i)}$: $p_i = (len(g^{(i)}) - n_{term}^{(i)})/len(g^{(i)})$. Higher values of $p_i$ indicate that a greater proportion of the trajectory $g^{(i)}$ belongs to intermediate process steps and is therefore assigned non-trivial step-level reward.

**Experimental Setup.** We trained two DeepSeek-R1-Distill-Qwen-1.5B models (DeepSeek-AI, 2025) on the OpenRS (Dang & Ngo, 2025) dataset using the standard GRPO algorithm and objective of Equation 2. We selected 125 OpenRS examples at random to serve as a validation set.

The first model trained for 1675 steps with a group size of six and a learn rate of $6 \times 10^{-6}$. The second was trained with a group size of 36 and a learn rate of $10^{-6}$ for 275 steps (due to the larger group size). Both models were trained with a maximum new token limit of 4096, a batch size of four, and a temperature of 0.75. Additional training details are located in Appendix B.

**Results.** Figure 2 shows that both path depth and intermediate proportion increase drastically as validation reward saturates, for group sizes of six and 36. These results are supported by Yu et al. (2025), who find that entropy decreases sharply as GRPO training progresses: this indicates that increasingly rich PRM-inducing structures arise as the model converges on a locally optimal policy.

In addition, found that only twelve of 6,700 $\mathcal{B}(\mathbb{G})$ structures were trivial with a group size of six ($\sim$0.2%). With a group size of 36, zero trivial $\mathcal{B}(\mathbb{G})$ structures arose out of the 1,100 generated groups. Examples of non-trivial $\mathcal{B}(\mathbb{G})$ structures from this experiment are given in Appendix C.

In conjunction with the theoretical analysis of Section 3.1, the results of this experiment demonstrate that GRPO induces a non-trivial PRM under real-world conditions. In Section 4, we show that this induced PRM carries a serious flaw that is detrimental to RL training, and propose a minor modification to the GRPO algorithm to mitigate this shortcoming.

## 4    PROPOSED APPROACH: $\lambda$-GRPO

Viewing the GRPO objective in terms of process set partitions $\mathbb{X}_t$ (see Equation 6), we note that the contribution of each trajectory $g^{(i)} \in \mathbb{G}$ to the loss at index $t$ is identical to that of all other trajectories in the process set $\lambda^{(i,t)}$ (where $\hat{P}_t(\lambda)$, $\hat{D}_t(\lambda)$, and $\hat{A}(\lambda)$ are defined as in Equation 5):

$$
\frac{1}{\sum_{g^{(i)} \in \mathbb{G}} len(g^{(i)})} \cdot \sum_{g^{(i)} \in \mathbb{G}} \sum_{t=0}^{len(g^{(i)})-1} (P_{i,t} \cdot a_i) - D_{i,t} = \frac{1}{\sum_{g^{(i)} \in \mathbb{G}} len(g^{(i)})} \cdot \sum_{t=0}^{t_{max}-1} \sum_{\lambda \in \mathbb{X}_t} \sum_{g^{(i)} \in \lambda} (P_{i,t} \cdot A_{i,t}) - D_{i,t}
$$
$$
= \frac{1}{\sum_{g^{(i)} \in \mathbb{G}} len(g^{(i)})} \cdot \sum_{t=0}^{t_{max}-1} \sum_{\lambda \in \mathbb{X}_t} |\lambda| \cdot ((\hat{P}_t(\lambda) \cdot \hat{A}(\lambda)) - \hat{D}_t(\lambda))
$$
(7)

The contribution of each process set $\lambda \in \mathcal{B}(\mathbb{G})$ to the overall loss, $\hat{P}_t(\lambda) \cdot \hat{A}(\lambda) - \hat{D}_t(\lambda)$, is scaled by $|\lambda|$: this carries the danger of harming exploration (for $\hat{A}(\lambda) < 0$) *and* exploitation (for $\hat{A}(\lambda) > 0$). Consider some process set $\lambda$ with $|\lambda| \gg 1$. If $\hat{A}(\lambda) > 0$, then the increase in probability assigned to the process step corresponding to $\lambda$ by $\pi_\theta$ under GRPO is compounded by a factor of $|\lambda|$, decreasing the likelihood of exploring process steps that are dissimilar from $\lambda$ in subsequent training episodes.

Conversely, if $\hat{A}(\lambda) < 0$, then the *decrease* in probability assigned to $\lambda$ under GRPO is compounded by a factor of $|\lambda|$, decreasing the likelihood of exploiting high-reward trajectories in $\lambda$. To illustrate, consider the group $\mathbb{G}$ in Figure 1, assume $r_1 = r_2 = r_6 = 0.5$, $r_4 = r_5 = 0$, $r_3 = 1$, and let $\lambda = \{g^{(3)}, g^{(4)}, g^{(5)}\}$. Then $\hat{A}(\lambda) = -0.22$: despite the fact that $g^{(3)}$ has the highest reward, the probability of the sub-trajectory $g_{:4}^{(3)}$ is *decreased* under the GRPO objective, thereby decreasing the overall likelihood of generating the completion $g^{(3)}$. The term $|\lambda|$ in Equation 7 then scales this decrease in probability by a factor of three.

We propose scaling the token-level loss for $g_t^{(i)}$ by $|\lambda^{(i,t)}|^{-1}$ ($\lambda$-GRPO; Equation 8): this has the effect of canceling out the term $|\lambda|$ in Equation 7, so that each process set contributes equally to the loss at index $t$.

$$
L_{\lambda\text{-GRPO}}(\mathbb{G}) = \frac{1}{\sum_{g^{(i)} \in \mathbb{G}} len(g^{(i)})} \cdot \sum_{g^{(i)} \in \mathbb{G}} \sum_{t=0}^{len(g^{(i)})-1} \frac{(P_{i,t} \cdot a_i) - D_{i,t}}{|\lambda^{(i,t)}|}
$$
$$
= \frac{1}{\sum_{g^{(i)} \in \mathbb{G}} len(g^{(i)})} \cdot \sum_{t=0}^{t_{max}-1} \sum_{\lambda \in \mathbb{X}_t} (\hat{P}_t(\lambda) \cdot \hat{A}(\lambda)) - \hat{D}_t(\lambda)
$$
(8)

## 5    EXPERIMENTS

To evaluate our proposed approach, we trained DeepSeek-R1-Distill-Qwen-1.5B and Llama-3.2-1B-Instruct[4] with the $\lambda$-GRPO (Equation 8) objective on the OpenRS dataset of Section 3.2, and compared them to standard GRPO (Equation 2a) models trained with an identical setup. All models were evaluated on an OpenRS validation set and five downstream reasoning benchmarks (see Section 5.1).

---

[4]https://huggingface.co/meta-llama/Llama-3.2-1B-Instruct

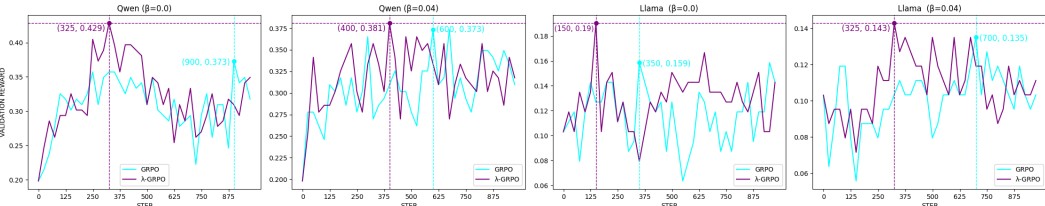

Figure 3: Models' validation accuracy across training steps. Peak accuracy is highlighted by vertical, dashed lines.

| Model | $\beta$ | Version | AIME24 | MATH-500 | AMC23 | Minerva | OB | Avg. |
|---|---|---|---|---|---|---|---|---|
| Qwen | — | Base | $0.2000_{\pm 0.0743}$ | $0.8300_{\pm 0.0168}$ | $0.7500_{\pm 0.0693}$ | $\mathbf{0.2978}_{\pm 0.0278}$ | $0.5096_{\pm 0.0193}$ | 0.5175 |
| | 0.0 | GRPO | $0.3333_{\pm 0.0875}$ | $0.7660_{\pm 0.0190}$ | $0.6250_{\pm 0.0775}$ | $0.2500_{\pm 0.0263}$ | $0.4444_{\pm 0.0191}$ | 0.4837 |
| | | $\lambda$-GRPO (ours) | $0.3667_{\pm 0.0895}$ | $0.8460_{\pm 0.0162}$ | $0.7500_{\pm 0.0693}$ | $0.2904_{\pm 0.0276}$ | $0.5348_{\pm 0.0192}$ | 0.5576 |
| | 0.04 | GRPO | $0.3000_{\pm 0.0851}$ | $\mathbf{0.8660}_{\pm 0.0152}$ | $0.7500_{\pm 0.0693}$ | $0.2610_{\pm 0.0267}$ | $0.5200_{\pm 0.0192}$ | 0.5394 |
| | | $\lambda$-GRPO (ours) | $\mathbf{0.4000}_{\pm 0.0910}$ | $0.8340_{\pm 0.0167}$ | $\mathbf{0.8000}_{\pm 0.0641}$ | $0.2978_{\pm 0.0278}$ | $\mathbf{0.5378}_{\pm 0.0192}$ | $\mathbf{0.5739}$ |
| Llama | — | Base | $0.0000_{\pm 0.0000}$ | $0.2280_{\pm 0.0188}$ | $0.0750_{\pm 0.0422}$ | $0.0478_{\pm 0.0130}$ | $0.0563_{\pm 0.0089}$ | 0.0814 |
| | 0.0 | GRPO | $0.0000_{\pm 0.0000}$ | $0.2300_{\pm 0.0188}$ | $0.0750_{\pm 0.0422}$ | $0.0551_{\pm 0.0130}$ | $0.0607_{\pm 0.0089}$ | 0.0842 |
| | | $\lambda$-GRPO (ours) | $0.0000_{\pm 0.0000}$ | $0.2620_{\pm 0.0197}$ | $0.1250_{\pm 0.0530}$ | $0.0515_{\pm 0.0134}$ | $0.0622_{\pm 0.0092}$ | 0.1001 |
| | 0.04 | GRPO | $0.0000_{\pm 0.0000}$ | $0.2180_{\pm 0.0185}$ | $0.1750_{\pm 0.0608}$ | $0.0515_{\pm 0.0134}$ | $0.0533_{\pm 0.0087}$ | 0.0996 |
| | | $\lambda$-GRPO (ours) | $0.0333_{\pm 0.0333}$ | $0.2560_{\pm 0.0195}$ | $0.0750_{\pm 0.0422}$ | $0.0735_{\pm 0.0159}$ | $0.0489_{\pm 0.0083}$ | 0.0973 |

Table 1: Exact-match accuracy for the base and GRPO-/$\lambda$-GRPO-trained Llama and Qwen models on downstream reasoning datasets (OB = OlympiadBench). The best results in each column are indicated in **bold**, and the best results within each model type (i.e. Llama or Qwen) are underlined. Confidence intervals are subscripted. For each $\lambda$-GRPO-trained model, results are given in green if it outperforms its GRPO-trained counterpart and the base model; yellow if it outperforms only its GRPO-trained counterpart; orange if it only improves over the base model; and red if it fails to outperform either model (see Table 2 in the Appendix for exact differences).

## 5.1 EXPERIMENTAL SETUP

All models were trained for 1,000 steps with a group size of six, a batch size of four, a maximum of 4,096 new tokens, and a temperature of 0.75. We conducted two sets of trials across the two models (for a total of four trials): in the first, we set the KL coefficient $\beta = 0.0$, and in the second $\beta = 0.04$. The Qwen models were trained with a learn rate of $10^{-6}$; the Llama models were trained with a learn rate of $5 \times 10^{-7}$ for the $\beta = 0.0$ trial and $10^{-7}$ for $\beta = 0.04$ (as training was highly unstable with higher learn rates for Llama). Additional training details are located in Appendix B.

We evaluated the models on the AIME24[5], MATH-500 (Hendrycks et al., 2021; Lightman et al., 2024), AMC23[6], Minerva (Lewkowycz et al., 2022), and OlympiadBench (He et al., 2024) benchmarks, using the LightEval framework Habib et al. (2023); Dang & Ngo (2025). All models were evaluated at the checkpoint corresponding to the step at which they achieved maximum validation accuracy. As in the experiment in Section 3.2, we withheld 125 examples as a validation set.

## 5.2 RESULTS AND DISCUSSION

All four $\lambda$-GRPO models reach a higher validation accuracy in fewer steps than their GRPO counterparts (see Figure 3): on average, $\lambda$-GRPO represents a more than 10% increase over the standard GRPO validation accuracy—in less than half of the number of training steps.

This increase in validation accuracy corresponds to improved performance on downstream reasoning tasks (see Table 1). In total, the $\lambda$-GRPO models outperform standard GRPO on 15/20 cells (excluding average performance) in Table 1, and they improve over the base Llama/Qwen models on 14/20 cells. Only the Llama $\lambda$-GRPO model with $\beta = 0.04$ failed to outperform its GRPO counterpart on average downstream performance—this model still outperformed standard GRPO on a majority (3/5) of the tasks.

---

[5]https://huggingface.co/datasets/AI-MO/aimo-validation-aime
[6]https://huggingface.co/datasets/AI-MO/aimo-validation-amc

In addition, these performance gains result in effectively zero training slowdown: as we are merely *detecting* $\mathcal{B}(\mathbb{G})$ structures that occur during training—rather than *generating* them (e.g. Kazemnejad et al., 2025; Hou et al., 2025; Yang et al., 2025a)—the added computational cost from $\lambda$-GRPO (vs. standard GRPO) is negligible.

## 6 RELATED WORK

**Monte Carlo Sampling for Finer-Grained Rewards.**   As discussed in Section 2.2, the labor-intensive human annotation required to obtain step-level rewards for PRM training has driven the development of heuristic methods for PRMs and PRM-like finer-grained reward signals, in particular those based on Monte Carlo estimation. Kazemnejad et al. (2025) replace the critic model in PPO with Monte Carlo estimation: multiple completions are sampled for each step, and the value for that step is derived from their mean outcome reward. On the other hand, Wang et al. (2024) train a neural PRM, using Monte Carlo estimation in a similar manner to Kazemnejad et al. (2025) in order to obtain step-level training rewards, and thereby avoid the need for costly human annotation.

Xie et al. (2024) generate step-level preference data for Direct Preference Optimization (DPO; Rafailov et al., 2023) training via Monte Carlo Tree Search and outcome-level rewards: sequences of overlapping trajectories are obtained by forking trajectories at defined split points to construct tree structures similar to the $\mathcal{B}(G)$ trees introduced in Section 3.1. The daughter nodes with the highest and lowest mean reward are then selected as the preferred and dispreferred (respectively) sequences for step-level DPO. Similarly, Hou et al. (2025) construct $\mathcal{B}(G)$-like trees by splitting generated trajectories at high-entropy tokens to create multiple completions to the same initial trajectory prefix. Subtrajectory rewards are then derived from the mean reward of the corresponding node's daughters in the tree structure. Yang et al. (2025a) employ an analogous approach to generate step-level rewards for GRPO training. Unlike standard GRPO, however, the advantages for each node (step) are computed relative to the rewards of node's sisters in the tree structure, rather than the entire group.

These methods are orthogonal to our approach: they apply Monte Carlo estimation to explicitly construct step-level reward signals from outcome-level rewards, while we leverage the implicit step-level rewards already present in standard GRPO.

**PRMs with GRPO.**   Aside from Yang et al. (2025a), GRPO has been employed with PRMs in Shao et al. (2024) and Feng et al. (2025). Shao et al. (2024) modify the advantage computation of GRPO to account for step-level rewards: normalized rewards are computed relative to all step-level rewards of all trajectories in the group, and the advantage for each step is the sum of the normalized reward of each subsequent step in its trajectory. Feng et al. (2025) construct a two-level variant of GRPO, in which standard, trajectory-level GRPO advantage is combined with a step-level GRPO advantage, and step-level groups are dynamically computed according to the similarity of steps across trajectories.

Our results in Sections 3 and 5 call into question the necessity of adapting GRPO to step-level rewards, given the rich step-level reward signal already present in the outcome-level variant of the algorithm.

**Connections between PRMs and Outcome-level Reward.**   (Rafailov et al., 2024) prove that DPO can learn any token-level reward function—expressed as the difference in conditional log probability between the policy and reference models—given an appropriate training dataset. In contrast, we prove that GRPO with a given outcome reward function $r$ is equivalent to a PRM-sensitive RL algorithm with a PRM whose process rewards are given by an on-policy Monte Carlo estimate of the expected reward under $r$ for the process step in question.

## 7 CONCLUSION

In this paper, we demonstrated both theoretically and empirically that the standard GRPO algorithm induces a PRM that derives step-level rewards via Monte Carlo estimation. We then showed that this hidden PRM is faulty, and as a result is potentially detrimental to exploration and exploitation. To mitigate this flaw, we introduced a process-step-aware scaling factor to GRPO to derive the $\lambda$-GRPO

objective. Models trained with $\lambda$-GRPO reach higher validation accuracy faster than with standard GRPO, and achieve improved performance on downstream reasoning tasks.

Our results indicate that it is possible to leverage the existing PRM structure inherent in the outcome-based GRPO algorithm, rather than employing costly, explicitly-defined PRMs. The limitations of this work are discussed in Appendix A.

## REPRODUCIBILITY STATEMENT

We provide the complete proof of Theorem 1 in the main body of the paper. All settings and hyper-parameters for the experiments conducted in this work are given in Sections 3.2/ 5.1 and Appendix B. We will make all relevant code available upon acceptance.

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

| Model | $\beta$ | Version | AIME24 | MATH-500 | AMC23 | Minerva | OB | Avg. |
|-------|------|---------|---------|----------|--------|---------|---------|---------|
| Qwen | 0.0 | Base | +0.1667 | +0.0160 | 0.0000 | -0.0074 | +0.0252 | +0.0401 |
| | | GRPO | +0.0334 | +0.0800 | +0.1250 | +0.0404 | +0.0904 | +0.0738 |
| | 0.04 | Base | +0.2000 | +0.0040 | +0.0500 | 0.0000 | +0.0282 | +0.0564 |
| | | GRPO | +0.1000 | -0.0320 | +0.0500 | +0.0368 | +0.0178 | +0.0345 |
| Llama | 0.0 | Base | 0.0000 | +0.0340 | +0.0500 | +0.0037 | +0.0059 | +0.0187 |
| | | GRPO | 0.0000 | +0.0320 | +0.0500 | -0.0036 | +0.0015 | +0.0160 |
| | 0.04 | Base | +0.0333 | +0.0280 | 0.0000 | +0.0257 | -0.0074 | +0.0159 |
| | | GRPO | +0.0333 | +0.0380 | -0.1000 | +0.0220 | -0.0044 | -0.0022 |

Table 2: Difference in accuracy between the $\lambda$-GRPO-trained models, and their corresponding base models and standard-GRPO-trained counterparts. Positive differences (i.e. $\lambda$-GRPO outperforms the comparison model) are highlighted in green; negative differences (i.e. the comparison model outperforms $\lambda$-GRPO) are highlighted in red. For example, the top-most entry in the AIME24 column indicates that the $\lambda$-GRPO Qwen model with $\beta = 0.0$ outperformed the base DeepSeek-R1-Distill-Qwen-1.5B by 0.1667 on the AIME24 benchmark.

# A  LIMITATIONS

Due to computational resource constraints, we were only able to conduct the experiments in Sections 3.2 and 5 with relatively small models: 1.5 billion (Qwen) and 1 billion (Llama) parameters. Similarly, we only use one dataset for RL training in both experiments—although OpenRS is a combination of the s1 (Muennighoff et al., 2025) and DeepScaleR Luo et al. (2025) datasets. Future work should extend our findings regarding the non-triviality of the GRPO-induced PRM and the effectiveness of $\lambda$-GRPO to larger models and more diverse (training) datasets.

Finally, the objective of this work is to expose the PRM induced by the GRPO algorithm, and to highlight the deficiencies of that PRM as described in Section 4. To that end, our proposed $\lambda$-GRPO method does not actually remedy the anti-exploitation effect of the GRPO-induced PRM—it merely lessens its impact. In future work, we intend to investigate more extensive modifications to the GRPO algorithm, with the goal of entirely solving the problems laid out in Section 4.

# B  EXPERIMENTAL SETUP

All experiments were conducted on a single NVIDIA H100 GPU. We trained all models with 24 gradient accumulation steps per step and a generation batch size of 6. The models were evaluated on the validation split every 25 training steps.

We additionally hard-coded the generation procedure to halt after "\boxed{...}" was detected: this was to prevent the model from generating multiple boxed answers for a single prompt.

# C  $\mathcal{B}(\mathbb{G})$  STRUCTURE EXAMPLES

The following (Figures 4, 5, 6) represent $\mathcal{B}(\mathbb{G})$ structures on groups generated during the group size 6 trial of the experiment in Section 3.2. A full trajectory is reconstructed by tracing the unique path from the root to a terminal node. The root (red) corresponds to the prompt/query $q$. Terminal nodes (yellow) denote singleton process steps $\{g^{(i)}\}$; each non-terminal node $\lambda$ (white; including the root) denotes the process step corresponding to the set of all terminal nodes dominated by $\lambda$. For the sake of presentation, overlong terminal steps are truncated with "...".

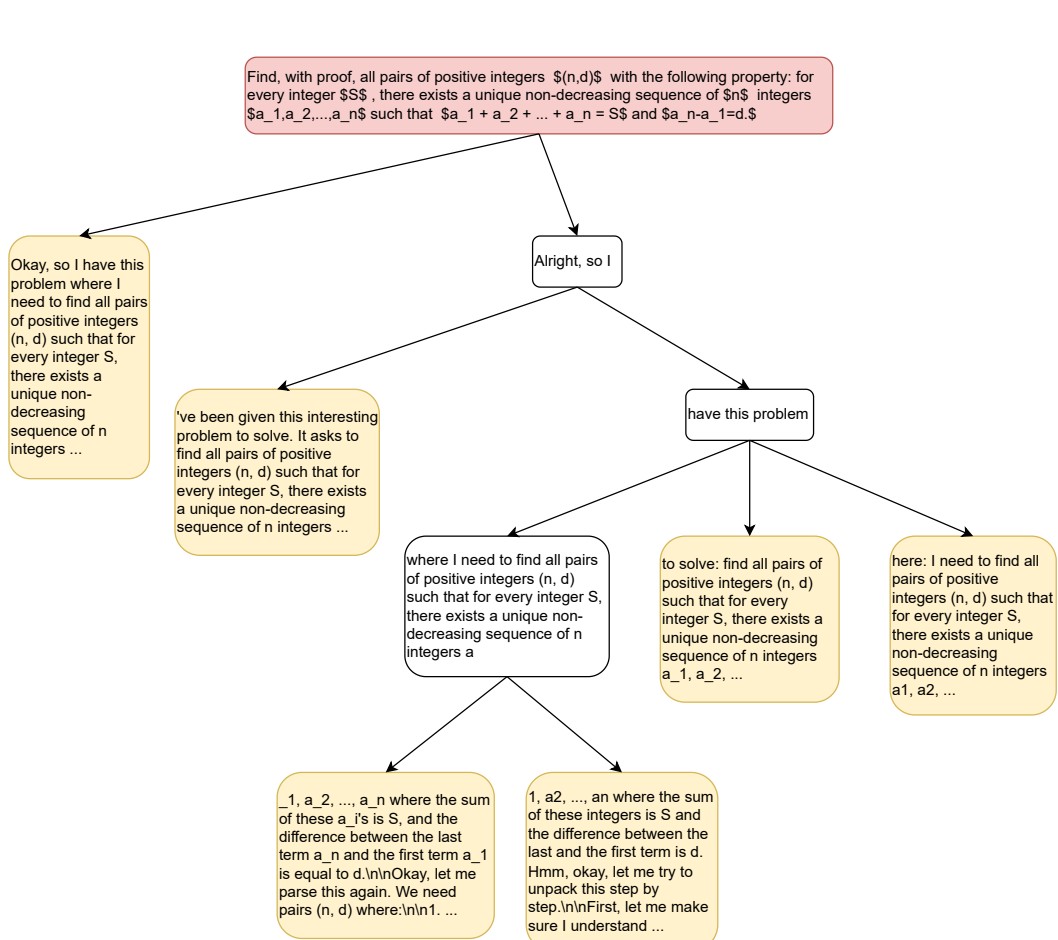

Figure 4: $\mathcal{B}(\mathbb{G})$ structure from step 1 (see the beginning of Appendix C for additional details).

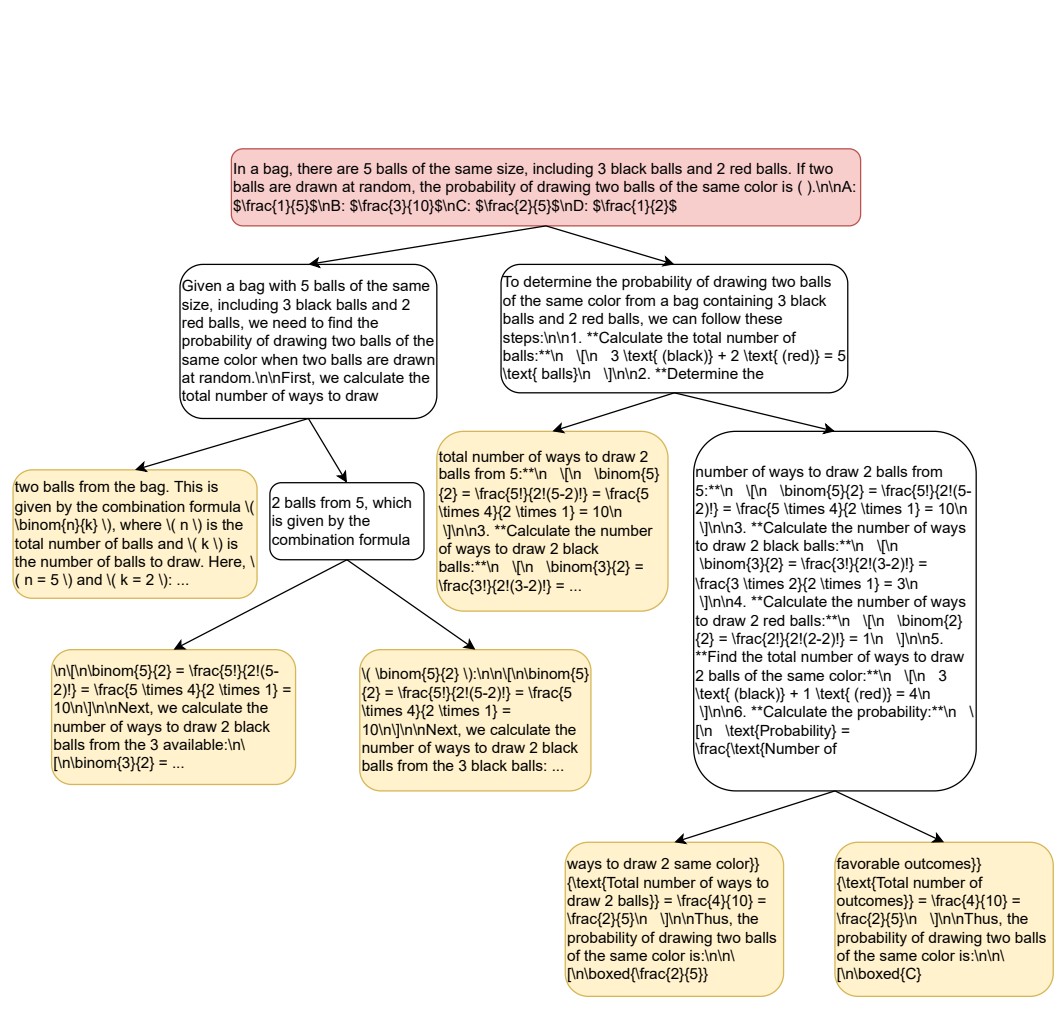

Figure 5: $\mathcal{B}(\mathbb{G})$ structure from step 1001 (see the beginning of Appendix C for additional details).

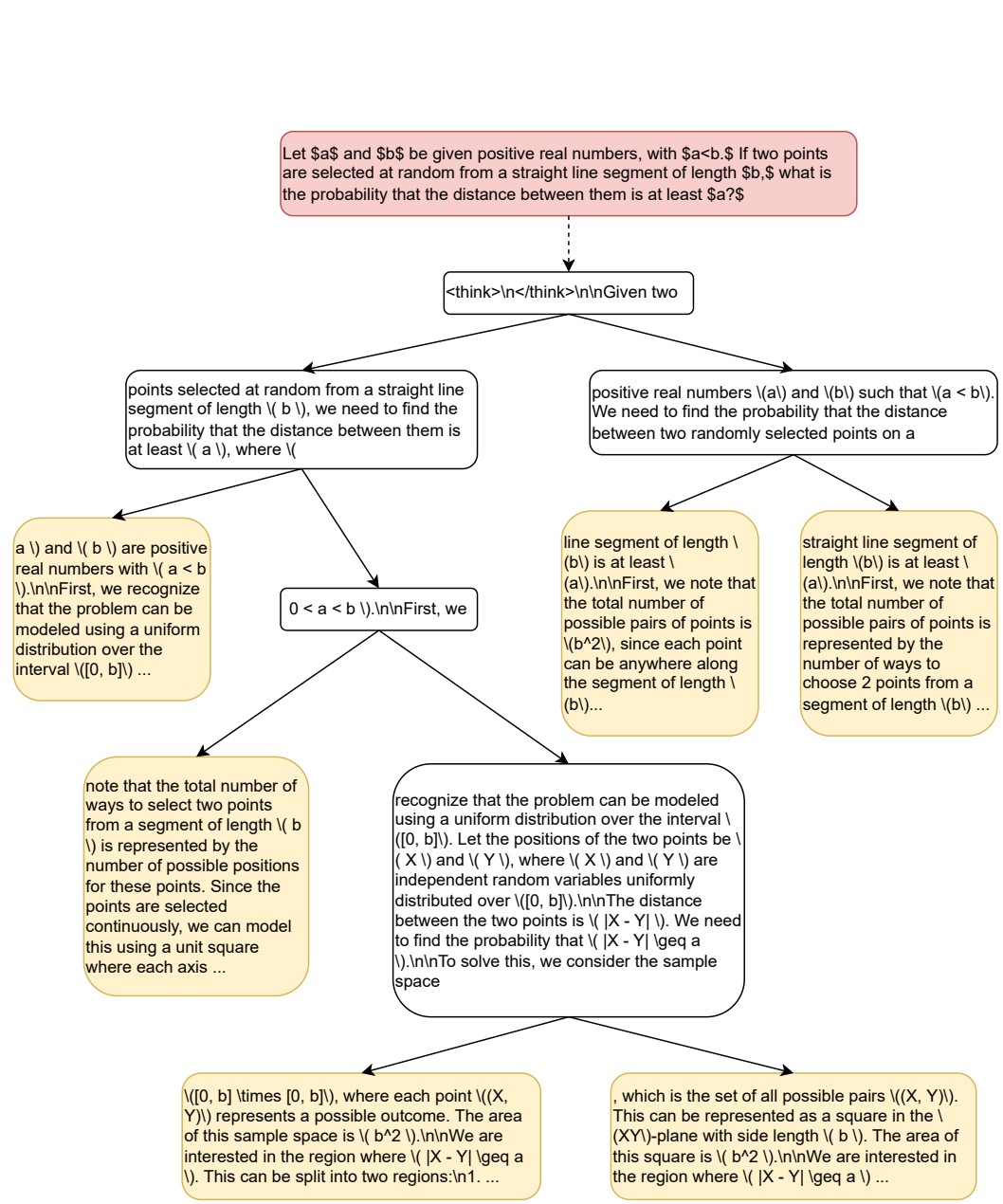

Figure 6: $\mathcal{B}(\mathbb{G})$ structure from step 1673 (see the beginning of Appendix C for additional details).

