# OpenReview forum: "GRPO is Secretly a Process Reward Model"
_ICLR.cc/2026/Conference — Submitted to ICLR 2026_

### Official Review · Reviewer_Tw6G · 2025-10-16

**Soundness:** 3
**Presentation:** 3
**Contribution:** 3
**Rating:** 6
**Confidence:** 3

**Summary:**

The paper connects the GRPO based on outcome verifiable reward to a similar GRPO algorithm trained with process reward. Meanwhile, it adds a process-number-aware factor to rescale the original GRPO loss, which is validated by their empirical studies.

**Strengths:**

- The connection between the outcome-level GRPO and process-level GRPO appears novel.

- The design of the rescaling factor is well-motivated and intuitive.

**Weaknesses:**

> Title

GRPO is an algorithm, PRM is a model. The main arguments of the paper is more like that GRPO can also be regarded as a PRM-trained RL algorithm.

> Definition of process reward

The process reward defined in Equation 3 is quite different from current PRM formulation. Actually, it is more like a value function (the total future reward a policy can expect from being in a particular state) than a reward function.

> Larger-scale Traning

The current experiments are conducted on 1.5B and 1B models, would this method remain effective for larger LLMs?

>Typos
- L136: ReACT-style prompting Yao et al. (2023) ->  ReACT-style prompting (Yao et al., 2023)
- L162: λ ∈ -> λ ⊆

**Questions:**

> Notations

In L174, the notation {g(4), g(5)} (follow-on similar notations too) is obscure. Please directly refer here to figure 1 (right) and add some demonstration for clarification.

>Diversity & Entropy

Since the \lambda-GRPO has a more conservative gradient descent, would there remain or ameliorate a sharp entropy decrease during the early training stage? Would \lambda-GRPO benefit the generative diversity?

---

> ### Author Response · Authors · 2025-11-12
>
> Thank you for your time and review. Here are our responses to your questions/concerns:
>
> > GRPO is an algorithm, PRM is a model. The main arguments of the paper is more like that GRPO can also be regarded as a PRM-trained RL algorithm.
>
> Yes, we agree that the title is somewhat imprecise. We will either change it or make this more clear in the abstract/introduction when we update the paper.
>
> > The process reward defined in Equation 3 is quite different from current PRM formulation. Actually, it is more like a value function (the total future reward a policy can expect from being in a particular state) than a reward function.
>
> The PRM defined in Section 3.1 conforms to our definition of a PRM given on L102-106. In addition, we have two counterexamples in our related work section:
>
> Wang et al. (2024) use Monte Carlo sampled estimates of expected outcome reward to train a PRM without human-annotated labels (L444-447): this is effectively training the PRM to predict expected future reward.
>
> Hou et al. (2025) use on-policy Monte Carlo sampled estimates of expected outcome reward to directly obtain step-level rewards (L452-456): these are directly plugged into PPO as process rewards, not values.
>
> > L162: $\lambda\in$ -> $\lambda\subseteq$
>
> This is not a typo: $\mathcal{B}(\mathbb{G})$ is a set of sets (L156), so an element $\lambda$ of $\mathcal{B}(\mathbb{G})$ is itself a set of trajectories.
>
> > In L174, the notation {g(4), g(5)} (follow-on similar notations too) is obscure. Please directly refer here to figure 1 (right) and add some demonstration for clarification.
>
> This is simply set notation: {$g^{(4)},g^{(5)}$} is just the set containing $g^{(4)}$ and $g^{(5)}$. If the reviewer will elaborate as to which aspect of this notation is obscure, we will correct this when we update the paper.
>
> > Since the $\lambda$-GRPO has a more conservative gradient descent, would there remain or ameliorate a sharp entropy decrease during the early training stage? Would $\lambda$-GRPO benefit the generative diversity?
>
> We actually did investigate this, and found no significant differences in entropy between models trained with $\lambda$-GRPO vs standard GRPO. We will certainly include these negative results if the reviewers believe it would improve the overall paper.

---

### Official Review · Reviewer_BrTV · 2025-10-25

**Soundness:** 2
**Presentation:** 1
**Contribution:** 2
**Rating:** 2
**Confidence:** 2

**Summary:**

The paper theoretically discusses how GRPO induces a Monte-Carlo-based PRM under certain mild assumptions, and proposes a fix on the GRPO loss which leads to better normalization. Experiments are conducted on several models, which show that the proposed method outperforms standard GRPO in terms of highest performance and efficiency.

**Strengths:**

This work provides valuable theoretical and empirical insights into GRPO's implicit process reward modeling. The finding that GRPO's built-in PRM can obviate the need for expensive explicit PRMs may be impactful to the broader community.

**Weaknesses:**

1. I believe the clarity of the paper needs to be vastly improved. I'm quite familiar with the basic setup of PRM methods but still find it uneasy to understand the core point delivered in the. work. For instance, Section 3 is titled "THE PRM IN THE MACHINE". What does this mean?   Also, I cannot find a precise definition of the underlying PRM in any equation of the paper. What is the relationship between the explicit ORM used in typical training and the PRM proposed method? We cannot prove anything without a strict and proper mathematical definition of a PRM.

2. Existing works already builds some connection between likelihood-ratios and PRMs,  for instance the relationship between PRM and DPO is discussed in earlier works such as  From  to : Your Language Model is Secretly a Q-Function.  What is the relationship between these works?

3. I do not see clear improvement from the validation plots provided. The highest peak result does not stably maintain.  and the improvement trends for GRPO and fixed-GRPO methods are similar.

**Questions:**

None

---

> ### Author Response · Authors · 2025-11-13
>
> Thank you for your time and review. Here are our responses to your questions/concerns:
>
> > Section 3 is titled "THE PRM IN THE MACHINE". What does this mean?
>
> This is a reference to the “Ghost in the Machine”, and refers to the fact that GRPO with outcome-level rewards behaves like (is equivalent to) a PRM-aware RL algorithm equipped with a PRM (see L044-048, L211-212)—it is a reference to the “hidden” PRM inside of GRPO. We will change this to a more precise title when we update the paper.
>
> > I cannot find a precise definition of the underlying PRM in any equation of the paper
>
> We precisely define the PRM on L190-194. Concretely, we define the token-level reward $R_{i,t}$: this is a process-step-level reward—the same reward is assigned to all tokens within a given process step. $R_{i,t}$ is defined on L191 to be $\hat{R}(\lambda^{(i,t)})$: $\hat{R}(\dots)$ is precisely defined in Equation 3, and $\lambda^{(i,t)}$ is precisely defined on L176-177.
>
> If the reviewer will point out the specific aspects of any of these definitions that they find imprecise or informal, we will gladly correct them in the paper.
>
> > What is the relationship between the explicit ORM used in typical training and the PRM proposed method?
>
> To be clear: we do not *propose* a PRM method. We prove that GRPO with an ORM behaves like (is equivalent to) a PRM + PRM-aware RL algorithm (see L044-048, L211-212).
>
> $\hat{R}(\lambda)$ is the mean outcome-level reward of all completions in the set $\lambda\in\mathcal{B}(\mathbb{G})$, as defined in Equation 3. $\lambda$ defines a process step, which corresponds to the prefix shared across all completions in $\lambda$ (see L156-166). The process reward for the $t^{th}$ token of the $i^{th}$ completion ($g_{t}^{(i)}$)  is $R_{i,t}=\hat{R}(\lambda^{(i,t)})$, where $\lambda^{(i,t)}$ is the unique set in $\mathcal{B}(\mathbb{G})$ that corresponds to the process step to which $g^{(i)}_t$ belongs.
>
> So the PRM assigns reward based on Monte-Carlo sampled outcome rewards of completions to the process step in question (see L044-048).
>
> > We cannot prove anything without a strict and proper mathematical definition of a PRM.
>
> Again, we formally define the PRM on L190-194. We then provide a formal definition of a PRM on L102-106, and show that the PRM defined on L190-194 meets this definition (L195-199).
>
> > Existing works already builds some connection between likelihood-ratios and PRMs, for instance the relationship between PRM and DPO is discussed in earlier works such as From to : Your Language Model is Secretly a Q-Function. What is the relationship between these works?
>
> This work is orthogonal to ours. Rafailov et al. (2024) prove that for any token-level reward function $r(x_{:t}, x_t)$ (that gives a reward for the token $x_t$ conditioned on the preceding context $x_{:t}$), DPO can learn a policy $\pi_\theta$ that *models* $r$: $r(x_{:t},x_t)\approx\beta\hspace{1mm}ln\hspace{1mm}\pi_\theta(x_t|x_{:t})-\beta\hspace{1mm}ln\hspace{1mm}\pi_{ref}(x_t|x_{:t})$ (for some constant $\beta$). So DPO “builds” a PRM from a given training dataset and model over the course of training, whose reward at the token $x_t$ is proportional to the difference in log probability between $\pi_\theta(x_t|x_{:t})$ and $\pi_{ref}(x_t|x_{:t})$.
>
> We prove that GRPO with a given outcome reward function $r$ is equivalent to a PRM-sensitive RL algorithm with a PRM whose process rewards are given by an on-policy Monte Carlo estimate of the expected reward under $r$ for the process step in question (see L045-048).
>
> > I do not see clear improvement from the validation plots provided. The highest peak result does not stably maintain. and the improvement trends for GRPO and fixed-GRPO methods are similar.
>
> We agree that this is not a substantial increase in validation performance. The main result of Section 5 is the nearly across-the-board improvement across five downstream tasks vs standard GRPO (Table 1).

---

### Official Review · Reviewer_8V6o · 2025-10-29

**Soundness:** 2
**Presentation:** 1
**Contribution:** 2
**Rating:** 2
**Confidence:** 4

**Summary:**

This paper theoretically proves that GRPO implicitly induces a non-trivial process reward model via Monte Carlo estimation on overlapping token prefixes in sampled trajectory groups. The authors identify a flaw and introduce λ-GRPO, normalizing the loss by process set size. The work questions the need for explicit PRMs in GRPO, advocating exploitation of its built-in structure for efficiency.

**Strengths:**

1. Provides a fresh theoretical lens on GRPO as a "secret" PRM, backed by a clean proof of objective equivalence (Theorem 1), potentially obviating costly explicit PRMs and advancing RLHF efficiency.
2. : λ-GRPO is a simple, low-overhead tweak (negligible compute cost), with clear diagnostics of exploration/exploitation flaws, offering immediate value for LLM training pipelines.

**Weaknesses:**

1. The paper is hard to follow. Writing could be improved. There are too many unnecessary symbols and unclear definitions.
	1. The usage of q, g to represent prompt and response is uncommon. g also contrasts with y in section 2.2
	2. the definition of $\lambda$ is very vague. I cant understand what $\lambda$ with superscrtpts represent.
	3. What does n_term mean? Isn't $e({g^i}) - s({g^i}) = len(g^i)$ according to the definition?
     This paper would benefit from a total rewrite.
2. Experiments use small models (1-1.5B parameters) on one dataset (OpenRS), ignoring larger scales or diverse domains; results may not extend to broader RL applications or non-math tasks.
3. λ-GRPO only dampens the non-uniformity issue via normalization but does not eliminate it as acknowledged in limitations.
4. Relies on specific GRPO variants (DAPO loss, μ=1 updates), deviating from standard implementations; lacks sensitivity analysis or other ablation studies
5. The small group size (6 in experiments) hardly enables prm learning in practice.

**Questions:**

1. On average, what is the ratio of overlaped prefix tokens in one group? I wonder this number would not be large enough for training a valid prm.

---

> ### Author Response · Authors · 2025-11-12
>
> Thank you for your review. Here are our detailed responses to your questions/concerns:
>
> > The paper is hard to follow. Writing could be improved. There are too many unnecessary symbols and unclear definitions
>
> We attempted to address this below, please let us know if anything else can be improved.
>
> > The usage of q, g to represent prompt and response is uncommon.
>
> q is used to represent the prompt in both the DeepSeekMath paper (which introduces GRPO) and the DAPO paper: we were following their convention. These papers use o to represent responses: we chose instead to use g to avoid potential confusion between o and the number 0. We will update this in the paper if it is a point of concern.
>
> > g also contrasts with y in section 2.2
>
> We agree, and will update the paper accordingly.
>
> > the definition of $\lambda$ is very vague.
>
> $\lambda$ is an element of $\mathcal{B}(\mathbb{G})$. This is to say that $\lambda$ is a subset of the group $\mathbb{G}$ such that all of the trajectories in $\lambda$ share a common prefix. This is precisely defined on L156-158. When defining $\mathcal{B}(\mathbb{G})$ on L156-158, we refer readers to Figure 1, in which this definition is illustrated. If the reviewer will let us know which aspect of these explanations was difficult to understand, we will gladly update them in the paper.
>
> > I can’t understand what $\lambda$ with superscripts represent
>
> As stated on L162-166, each $\lambda$ corresponds to a process step, which is the shared prefix that defines $\lambda$: the process step corresponding to $\lambda$ spans the $s(\lambda)^{th}$ to $e(\lambda)^{th}$ tokens of each trajectory in $\lambda$.
>
> In Figure 1, $g^{(3)}$, $g^{(4)}$, $g^{(5)}$ share a prefix spanning the first four tokens of those trajectories, so $e(${$g^{(3)}, g^{(4)}, g^{(5)}$}$)=4$. $g^{(4)}$ and $g^{(5)}$ further share a prefix spanning the first six tokens of those trajectories (a superstring of the prefix corresponding to {$g^{(3)}, g^{(4)}, g^{(5)}$}): $e(${$g^{(4)}, g^{(5)}$}$)=6$, and $s(${$g^{(4)}, g^{(5)}$}$)=4$, because $e(${$g^{(3)}, g^{(4)}, g^{(5)}$}$)=4$. $s(\lambda)$ and $e(\lambda)$ are defined on L168-171, and we give an example (illustrated in Figure 1) on L173-175.
>
> $\lambda^{(i,t)}$ is simply the unique $\lambda$ whose corresponding process step contains the $t^{th}$ token of the trajectory $g^{(i)}$: i.e. the unique $\lambda$ such that $g^{(i)}$ is a member of $\lambda$ and $s(\lambda)\leq t<e(\lambda)$. This definition of $\lambda^{(i,t)}$ is given on L176-178, and we give an example (illustrated in Figure 1) on L178-180.
>
> Again, if the reviewer will let us know which aspect of these explanations and definitions was difficult to understand, we will gladly update them in the paper.
>
> > What does n_term mean? Isn't $e(g^{i})-s(g^{i})=len(g^{i})$ according to the definition?
>
> Note that the definition of n_term on L302 is in fact $e(${$g^{i}$}$)-s(${$g^{i}$}$)$: e/s are taking *the singleton set containing* $g^{i}$ as arguments—i.e. the $\lambda$ that only contains $g^{i}$. By the definition of $s(\lambda)$ and $e(\lambda)$ on L168-171, $e(${$g^{i}$}$)=len(g^{i})$, and $s(${$g^{i}$}$)$ is the start of the span of tokens in $g^i$ that are not part of a prefix shared with any other trajectory. So $n^{(i)}_{term}=e(${$g^{i}$}$)-s(${$g^{i}$}$)$ is the number
> of tokens unique to $g^{i}$ (i.e. not part of a shared prefix / not overlapped), as stated on L303-304.
>
> > Experiments use small models (1-1.5B parameters) on one dataset (OpenRS), ignoring larger scales or diverse domains; results may not extend to broader RL applications or non-math tasks.
>
> We did not include larger models due to our limited compute budget. We will certainly attempt to include experiments using additional training/evaluation datasets, but we unfortunately cannot guarantee that these experiments will be completed before the end of the author response period.
>
> > Relies on specific GRPO variants (DAPO loss, $\mu=1$ updates)
>
> We believe that DAPO loss is in fact the current standard GRPO implementation: for example, it is the default loss function for the TRL GRPO trainer (see L079-080).
>
> We agree that the assumption of $\mu=1$ is a limitation of the current work. We have derived a PRM for $\mu>1$ (it is just less clean and intuitive than with $\mu=1$): we can include this theoretical analysis if the reviewers believe it would improve the overall paper.
>
> > The small group size (6 in experiments) hardly enables prm learning in practice.
>
> We respectfully disagree: our study in Section 3.2 quantitatively demonstrates that even with group size = 6, we have a large amount of non-trivial overlap within groups (see further below in this response). Also see Figures 4-6 in the Appendix
>
> > On average, what is the ratio of overlapped prefix tokens in one group?
>
> This is precisely what we study in our experiment in Section 3.2: “intermediate proportion” is the proportion of overlapped prefix tokens. See L302-306 and Figure 2.

---

### Author Response · Authors · 2025-12-03
**Summary of discussion and changes made**

The majority of the weaknesses noted by reviewers BrTV and 8V6o seem to stem from their perception of a lack of clarity in our paper. For example, reviewer 8V6o requests experiments that were already included in the original submission. It is additionally unclear to us how the low scores given by reviewer BrTV follow from the weaknesses listed.

While we are entirely open to editing the paper for clarity, we unfortunately did not receive any responses from the reviewers to our requests for elaboration, despite submitting our replies (Nov. 12) well before discussion was disabled (Nov. 28).

We did address those comments that we were able to without reviewer clarification: we re-named $y$ to $g$ in Section 2.2 in response to reviewer 8V6o; and renamed Section 3 to "GRPO's Hidden PRM" and included a discussion of Rafailov et al. (2024) in Section 6 in response to reviewer BrTV.

Additionally, we would like to point out that reviewer Tw6G did not mention any difficulty understanding the paper, and appears to have a solid understanding of the work, based on their constructive comments that we are looking forward to incorporating when possible (for example, we will change the title when permitted, and will include results with larger models once those experiments have completed).

---

### Meta-Review · Area_Chair_x9WC · 2026-01-06

**Summary:**

This paper offers an interesting theoretical perspective on GRPO, arguing that outcome-based GRPO implicitly induces a process reward model and proposing a simple normalization variant (λ-GRPO). The majority of concerns centered on clarity and presentation: two reviewers found the paper difficult to follow due to heavy notation, unclear definitions, and imprecise framing of what constitutes a PRM. Reviewers also questioned the conceptual positioning relative to prior work linking likelihood-ratio objectives and PRMs. Empirically, concerns were raised about limited scale and scope, reliance on small models and specific GRPO variants, and modest or unstable gains, which limit confidence in generality and practical impact.

**Reviewer Concerns:**

The rebuttal effectively addresses many technical correctness and definitional issues, but clarity, empirical generality, scalability, and strength of evidence remain outstanding and materially affect the paper’s readiness for acceptance.

**Reviewer Scores:**

Unfortunately, there is no discussion between authors and reviewers. If reviewers had been able to participate fully in the discussion, their score may be slightly increased, but still negative in average.

---

### Decision · Program_Chairs · 2026-01-26

Reject